# Preparation and Molecular Dynamic Simulation of Superfine CL−20/TNT Cocrystal Based on the Opposite Spray Method

**DOI:** 10.3390/ijms25179501

**Published:** 2024-08-31

**Authors:** Junming Yuan, Zhenyang Liu, Tao Han, Junyi Li, Peijiang Han, Jing Wang

**Affiliations:** School of Environmental and Safety Engineering, North University of China, Taiyuan 030051, China; sz202314035@st.nuc.edu.cn (Z.L.); sz202214026@st.nuc.edu.cn (T.H.); sz202214023@st.nuc.edu.cn (J.L.); s202314006@st.nuc.edu.cn (P.H.); sz202214083@st.nuc.edu.cn (J.W.)

**Keywords:** CL−20/TNT, cocrystal, opposite spray method, molecular dynamic simulation, pneumatic atomized droplets

## Abstract

In view of the current problems of slow crystallization rate, varying grain sizes, complex process conditions, and low safety in the preparation of CL−20/TNT cocrystal explosives in the laboratory, an opposite spray crystallization method is provided to quickly prepare ultrafine explosive cocrystal particles. CL−20/TNT cocrystal explosive was prepared using this method, and the obtained cocrystal samples were characterized by electron microscopy morphology, differential thermal analysis, infrared spectroscopy, and X-ray diffraction analysis. The effects of spray temperature, feed ratio, and preparation method on the formation of explosive cocrystal were studied, and the process conditions of the pneumatic atomization spray crystallization method were optimized. The crystal plane binding energy and molecular interaction forces between CL−20 and TNT were obtained through molecular dynamic simulation, and the optimal binding crystal plane and cocrystal mechanism were analyzed. The theoretical calculation temperature of the binding energy was preliminarily explored in relation to the preparation process temperature of cocrystal explosives. The mechanical sensitivity of ultrafine CL−20/TNT cocrystal samples was tested. The results showed that choosing acetone as the cosolvent, a spraying temperature of 30 °C, and a feeding ratio of 1:1 was beneficial for the formation and growth of cocrystal. The prepared CL−20/TNT cocrystal has a particle size of approximately 10 μm. The grain size is small, and the crystallization rate is fast. The impact and friction sensitivity of ultrafine CL−20/TNT cocrystal samples were significantly reduced. The experimental process conditions are simple and easy to control, and the safety of the preparation process is high, providing certain technical support for the preparation of high-quality cocrystal explosives.

## 1. Introduction

At present, many countries around the world have taken insensitive ammunition as the development direction of current ammunition and have conducted focused research on explosive formula and performance, charge structure, evaluation standards for explosive insensitivity, and experimental methods [1,2,3,4]. Research on the insensitivity and safety of the main explosive of ammunition loading can improve the safety and battlefield survival ability of ammunition, reduce the risk of explosion caused by accidental stimulation of weapons and ammunition in practical use, and reduce accidents of ammunition explosion caused by accidental impact and burning. The single-substance explosive CL−20 has high density, high energy, good chemical and thermal stability, and better comprehensive performance than octahydro-1,3,5,7-tetranitro-1,3,5,7-tetrazocine (HMX) and hexahydro-1,3,5-trinitro-1,3,5-triazine (RDX). It is widely used in high-performance explosives, new-generation propellants, and solid propellant formulations [5].

In order to meet the requirements of the development of high-energy insensitive explosives, explosive cocrystal technology combines two or more molecules into the same lattice through intermolecular interactions to develop new type of explosives with better comprehensive performance [6,7,8,9,10,11,12,13,14,15,16,17]. There are many methods to prepare cocrystal explosives, including the solution crystallization method, the solvent/nonsolvent method, the grinding method, and the spray drying method. The preparation of cocrystal explosives through various experimental methods provides a significant modification effect for energetic materials, which is conducive to expanding the research and application scope of energetic materials. However, cocrystal preparation technology still has defects such as crystallization rate, grain size, and experimental safety.

Zhang, X. et al. [7] proposed a new nitramine/energetic ionic salt cocrystal explosive, which contains CL−20 and 1-AMTN in a 1:1 molar ratio. Hu, Y. et al. [8] successfully prepared nano CL−20/TNT cocrystal explosives by mechanical ball milling with 0.38 mm grinding beads, with an average particle size of 119.5 nm and a spherical micromorphology. Xu, H. et al. [9] prepared the novel cocrystal explosive CL−20/TATB through a rapid nucleation solvent/nonsolvent process. Bolton, O. et al. [10] discovered and characterized an energetic–energetic cocrystal composed of a 1:1 molar ratio of the established explosives CL−20 and TNT. The results indicate that, compared to pure CL−20, an approximate doubling of the impact stability has been achieved. The solvent evaporation method [11,12,13,14,15] can control crystal morphology and size, but it takes a relatively long time and cannot use solvents with high solubility. The solvent/nonsolvent method [15,16,17,18] can achieve rapid cocrystal formation, but the crystal particle size is relatively large. The grinding method [18,19,20,21,22,23] is easy to operate. But there is a possibility of explosion during the process, which poses certain risks. The spray drying method [23,24,25] can shorten the crystal growth process and is conducive to the formation of ultrafine cocrystal particles. The disadvantage is that the process conditions are complex and difficult to control, and the safety is not high.

Many scholars have conducted extensive molecular dynamic simulations on the cocrystal structure of CL−20/TNT, as shown in Figure 1. Guo, D. et al. [26,27,28,29] investigated the thermal decomposition performance of the CL−20/TNT cocrystal. The results indicate that the cocrystal has a lower decomposition rate than CL−20 but a higher rate than TNT. These results confirm the expectation that cocrystallization is an effective method to reduce the sensitivity of energetic materials while maintaining high performance. In addition, molecular dynamic simulations were conducted to investigate the structure [30,31], intermolecular interactions [32,33,34], various kinds of sensitivity [35,36,37], thermal stability, mechanical properties [38,39], energetic performance, and detonation performance [40,41,42] of CL−20/TNT cocrystal and composite models under corresponding force fields. This provided theoretical support for the preparation and characterization of CL−20/TNT cocrystal experiments.

The commonly used experimental preparation methods for cocrystal explosives are still in the laboratory stage, with slow cocrystal speed, low production, and a significant gap from practical applications, making it difficult to achieve industrial mass production of cocrystal preparation. Due to the high sensitivity of explosives to stimuli such as heat, impact, and friction, which poses significant risks, in order to ensure the safety of the experimental process, cocrystals of explosives are generally prepared using a solution system instead of a non-solution system.

In response to the problems of complex process, long preparation time, low yield, and uneven particle size distribution in the current preparation of explosive cocrystal technology, the CL−20/TNT cocrystal was prepared based on the opposite spray method with pneumatic atomized droplets to rapidly produce an ultrafine explosive cocrystal. According to characterization by electron microscopy morphology, differential thermal analysis, infrared spectroscopy, and X-ray diffraction, it has been proven that this method can produce CL−20/TNT cocrystal explosives with good performance. Through molecular dynamics simulation, the binding energy between CL−20/TNT molecules was calculated, and the cocrystal mechanism was explored. By studying and analyzing the principles of spray technology and experimental influencing factors, certain theoretical support is provided for the effective preparation of stable cocrystal explosives.

## 2. Results

In order to confirm that the explosive crystal prepared by the opposite spray method with pneumatic atomized droplets is a CL−20/TNT cocrystal explosive, the appearance morphology, thermal decomposition characteristics, characteristic peak frequency, and diffraction peak of the explosive cocrystal were analyzed through characterization with electron microscopy scanning observation, DSC differential thermal analysis, infrared spectroscopy testing, and X−ray diffraction testing, respectively. The characterization results showed that the CL−20/TNT cocrystal explosive sample was obtained.

### 2.1. Characterization by Scanning Electron Microscopy

Scanning electron microscopy is an effective means of characterizing the morphology and size of cocrystal explosives. Scanning electron microscopy was used to observe and test the raw materials CL−20 and TNT and the products obtained from the experimental preparation. The scanning electron microscopy test results of several samples are shown in Figure 2.

### 2.2. Differential Thermal Testing Results

The thermal analysis method based on Differential scanning calorimetry (DSC) can be used to test and analyze the thermal performance and thermal decomposition process of samples. A DSC thermal analyzer was used to perform differential thermal testing on the raw materials CL−20 and TNT and the CL−20/TNT cocrystal products obtained from the experimental preparation. The DSC thermal analyzer was set with a starting temperature of 50 ℃ and an ending temperature of 350 ℃, and the heating rate was set to 10 ℃/min. After the heating was completed, the DSC thermal curve of the measured sample was obtained. The DSC thermal analysis test results of the CL−20, TNT, and CL−20/TNT test samples are shown in Figure 3. The intersection point between the tangent line (red dashed line) of the left side line of the thermal decomposition reaction peak and the horizontal axis of temperature is the initial decomposition temperature of the test sample.

### 2.3. DSC Testing Results of Explosive Cocrystal with Different Feeding Ratio

Choosing the appropriate feeding ratio is a key issue in preparing high-quality explosive cocrystals. According to existing public literature reports, the CL−20/TNT explosive co-solution currently used for preparing explosive cocrystals is mainly prepared in a 1:1 molar ratio. Due to the difficulty of using all explosives dissolved in solvents to form explosive cocrystal, the feeding ratio and the final cocrystal formation ratio may not be consistent during the preparation experiment. Therefore, it is necessary to consider the feeding ratio of the two components and create conditions that are conducive to the formation and growth of the cocrystal.

At room temperature, there is a slight difference in solubility between CL−20 and TNT explosives in acetone solvent, with TNT having a slightly higher solubility. An experimental method for preparing the CL−20/TNT cocrystal explosive with a feed ratio of 1:1 was adopted, ensuring that other conditions remained unchanged. The feed ratios of CL−20 and TNT were selected as 1:2, 1:3, and 1:4, respectively, for the cocrystal experiments. Finally, DSC testing was conducted on the prepared cocrystal samples to compare the effect on the explosive cocrystal of different feeding ratios. The most suitable ratio for preparing the CL−20/TNT cocrystal was given by the opposite spray method. The DSC test results of the CL−20/TNT explosive cocrystal with different feed ratios (1:2, 1:3, 1:4) are shown in Figure 4.

### 2.4. Infrared Spectrum Testing Results

Infrared spectroscopy analysis is a method of determining molecular structure and properties based on unique infrared spectra, which provides a certain reference for studying the molecular structure of samples. An infrared spectrometer was used to test and analyze the raw materials CL−20 and TNT and the products obtained from the experimental preparation. The infrared spectrum test results of several samples are shown in Figure 5.

### 2.5. X-ray Diffraction Testing Results

X-ray diffraction is a widely used method to determine the structure of samples by studying and analyzing the structures of different substances based on characteristic diffraction lines. CL−20, TNT, and their cocrystal explosive products were tested using an X-ray diffractometer. The diffraction test results of several samples are shown in Figure 6.

### 2.6. Simulated Results of Binding Energy between Molecular Crystal Planes

The different components of explosives form cocrystal structures through strong intermolecular interactions. Hydrogen bonding, as one of the most important intermolecular interactions, is a decisive factor in determining the possibility of cocrystal structure formation. Analysis of the strength of intermolecular interactions in different systems is of great significance for determining the feasibility of cocrystal formation. Binding energy is a characteristic parameter of the strength of intermolecular interactions between components, defined as the negative value of intermolecular interaction energy, which can well reflect the degree of mutual fusion between components [33] and can be expressed as:E_bind_ = −E_inter_ = −[E_total_ − (E_layer (1)_ + E_layer (2)_)](1)

In the formula, E_bind_ represents the binding energy; E_inter_ represents the intermolecular interaction energy; E_total_ represents the single-point energy obtained from the equilibrium structure; E_layer (1)_ is the first-layer single-point energy calculated when removing the second layer portion; E_layer (2)_ is the second-layer single-point energy calculated when removing the first layer portion. The energy calculation results of different cocrystal models are shown in Table 1, Table 2, Table 3, Table 4 and Table 5 below.

### 2.7. Calculated Results of Radial Distribution Function for CL−20/TNT Cocrystal

To investigate the mechanism of interaction between CL−20/TNT cocrystal structures, the radial distribution function was calculated through molecular dynamic simulation, and the calculation results are shown in Figure 7.

### 2.8. Testing Results of Mechanical Sensitivity

According to the characteristic drop height method specified in Method 601.2 of GJB-1997 “Test Methods for Explosives” [43], the impact sensitivity is tested with a charge of 30 ± 1 mg and a hammer mass of 2.00 ± 0.002 kg. At the same time, according to the explosion probability method specified in 602.1, friction sensitivity was tested with a swing angle of 90° and a gauge pressure of 3.92 MPa. The temperature range for mechanical sensitivity testing is 10–35 °C, with a relative humidity not exceeding 80%.

In the paper, the impact sensitivity of explosive cocrystal samples is characterized by a 50% drop height H_50_, while the explosion probability value is used as the magnitude of their friction sensitivity. The test results of the mechanical sensitivity of the CL−20/TNT explosive cocrystal are shown in Table 6.

## 3. Discussion

In order to confirm that the explosive crystal prepared by the opposite spray method with pneumatic atomized droplets is a CL−20/TNT cocrystal explosive, the appearance morphology, thermal decomposition characteristics, characteristic peak frequency, and diffraction peak of the explosive cocrystal were analyzed through characterization with electron microscopy scanning observation, DSC differential thermal analysis, infrared spectroscopy testing, and X-ray diffraction testing, respectively. The characterization results showed that the CL−20/TNT cocrystal explosive sample was obtained.

### 3.1. Characterization Analysis of Scanning Electron Microscopy

It can be seen from Figure 2a that most of the CL−20 grains exhibit rectangular block shapes under 500× times electron microscope magnification, with sizes ranging from 100 to 200 μm. A few grains exhibit irregular granular shapes, with sizes ranging from 20 to 50 μm. Figure 2b shows the morphology characterization results of TNT grains under 1000× electron microscope magnification. The characterization results show that the shape of TNT grains is mainly strip-like, with many defects, and some small grains are mixed on the surface. Figure 2c,d show the crystal morphology of CL−20/TNT cocrystal explosive grains under 5000× and 10,000× times electron microscope magnification, respectively. The characterization results in Figure 6c show that the CL−20/TNT grain size prepared by the opposite spray method with pneumatic atomized droplets is small, with an average size of only 1-2 μm. In addition, they were easy to unite into a sphere of about 20 μm. The reason why grains tend to agglomerate as they become smaller may be due to the reduction of grain size to near-nanometer level, which increases their surface free energy and surface activity, leading to easy mutual adsorption of grains. Another possible reason is that, after the collision of co-solution droplets with antisolvent water droplets, the grains are precipitated, which improves the surface wettability and enhances the bonding force between the particles, leading to easy aggregation of explosive cocrystals.

### 3.2. Differential Thermal Testing Analysis

It can be seen from Figure 3c that there is a melting endothermic peak and two decomposition exothermic peaks in the DSC thermal curve of the experimental product. The peak temperature of the melting endothermic process is 139.0 °C, and the two peak temperatures of the decomposition exothermic process are 223.8 °C and 252.1 °C, respectively. Compared with the thermal performance parameters of raw materials CL−20 and TNT, there is a significant difference in peak temperature, indicating that the product as a substance is different from a single component. According to reference [44], the melting point of CL−20/TNT cocrystal single crystals obtained by solvent evaporation method is 143.5 °C, and the thermal decomposition points are 222.6 °C and 250.1 °C, respectively. The thermal performance parameters of the product are basically consistent with them, indicating that they are the same substance. Therefore, this means that the CL−20/TNT explosive cocrystal product prepared by the opposite spray method with pneumatic atomized droplets can be preliminarily determined.

Changes in temperature have a certain impact on the transformation of CL−20 crystal forms. When the temperature is below 64 °C, the thermodynamics are the most stable at ε- CL−20. The product can be continuously heated for 6 weeks without transforming into other crystal forms, but when the temperature rises to above 74 °C, it will transform into γ-CL−20, and this transformation process is irreversible. When the temperature reaches around 164 °C, CL−20 undergoes complete crystal transformation [45]. According to the DSC curve analysis of Figure 3c, the CL−20/TNT cocrystal exhibits an endothermic peak at 139.0 °C. This is due to the destruction of the intermolecular interaction between the two components in the cocrystal structure and cocrystal transformation into the single components CL−20 and TNT. Continuing to heat up, there are two exothermic peaks in the DSC curve. The peak temperature should have been the decomposition temperature of the two single components, but there is a difference compared to the two single components. It may be that, after converting to a single component, the two components mix, which will have a certain impact on each other, resulting in different exothermic peak temperatures during the heating process compared to the single components.

From Figure 3a, the initial thermal decomposition temperature of the raw material CL−20 is about 233.3 °C, as indicated by the red dashed line, while Figure 7c shows that the initial thermal decomposition temperature of the CL−20/TNT cocrystal explosive is about 191.7 °C, as indicated by the red dashed line. This indicates that the initial decomposition temperature of the CL−20/TNT cocrystal explosive is about 41.6 °C lower than that of CL−20. In Figure 3a, it is shown that the raw material CL−20 releases a large amount of heat in a short period of time after starting to react and release heat at the initial decomposition temperature, with a maximum temperature of 244.2 °C, presenting a sharp triangle shape. Figure 7c shows the two peak temperatures exhibited by the thermal decomposition of the CL−20/TNT cocrystal explosive, and the exothermic area of the reaction is much larger than that of the raw material CL−20, indicating that the reaction heat of the cocrystal explosive is more significant.

### 3.3. The Influence of Feeding Ratio on Explosive Cocrystal

From the DSC curves of explosive cocrystals with different feed ratios of 1:1, 1:2, 1:3, and 1:4 in Figure 3c and Figure 4, the DSC curve with a feed ratio of 1:1 only has one endothermic peak, followed by two exothermic peaks. The thermal performance exhibited is basically consistent with the performance of CL−20/TNT explosive cocrystals reported in current public literature. However, when the feeding ratios were 1:2, 1:3, and 1:4, the prepared samples showed two endothermic peaks in the early stage. When the feeding ratio was 1:2, the prepared sample began to exhibit two melting points, with melting temperatures of 81.5 °C and 140.6 °C, respectively. Only one exothermic decomposition process has a peak temperature of 212.2 °C, and the first melting point is close to the melting point of TNT at 78.4 °C. This should be due to the increase in TNT dosage, with a portion of it precipitating as a single component. The second melting point is generated by cocrystal melting, and the thermal decomposition peak differs from that of CL−20/TNT cocrystals reported in the existing literature. When the feeding ratio was 1:3, the peak temperatures of the endothermic peaks of the prepared sample were 80.2 °C and 136.5 °C, with only one exothermic peak at 211.4 °C. There is an endothermic peak of TNT and an endothermic peak of the cocrystal, and the exothermic peak is different from that of cocrystal. When the feeding ratio was 1:4, the melting points of the prepared samples were 82.2 °C and 135.9 °C, with the former being more obvious. The thermal decomposition temperature points are 208.7 °C and 256.2 °C, respectively, and the changes are more significant compared to the CL−20/TNT cocrystal.

According to the DSC test results of the samples prepared with different feed ratios mentioned above, it can be concluded that changing the feed ratio during the preparation process will result in the production of a single component, which will affect the yield and quality of the cocrystal explosives. Using an inappropriate feeding ratio, due to the complex intermolecular interactions, it is even difficult to form cocrystal. From the test results, a 1:1 feeding ratio is more suitable for the CL−20/TNT cocrystal explosive prepared by the spray method, which is basically consistent with the research results reported in the current literature.

### 3.4. Testing Analysis of Infrared Spectrum

According to the basic principle of chemical spectral analysis, the vibration frequency of -NO_2_ group asymmetric stretching is found in the range of 1550 to 1500 cm^−1^. From Figure 5a,b, the asymmetric stretching vibration frequency of the -NO_2_ group in the pure component CL−20 occurs at an absorption peak of 1576 cm^−1^, while the absorption peaks of the -NO_2_ group in TNT occur at 1602 cm^−1^ and 1533 cm^−1^. Although the test values of the absorption characteristic peaks of the two components deviate slightly from the theoretical values, they are basically in good agreement. Unlike the characteristic peaks of the two pure components, it can be seen from Figure 9 that the absorption characteristic peak of the -NO_2_ group in the CL−20/TNT cocrystal structure is generated at the position of 1537 cm^−1^. Similarly, from spectroscopic theory, the frequency of symmetric stretching of the -NO_2_ group is about 1350 cm^−1^. However, from the spectral curve analysis in Figure 5, the frequencies of symmetric stretching of the -NO_2_ group in the two pure CL−20 and TNT sample components are at 1347 cm^−1^ and 1324 cm^−1^, respectively. The frequency of symmetric stretching of the -NO_2_ group in the cocrystal sample has an absorption characteristic peak at 1340 cm^−1^. In addition, the vibration frequency of the =C-H group is within the range of 3100 to 3000 cm^−1^. The absorption characteristic peaks of the pure component TNT appear at 3096 cm^−1^ and 3058 cm^−1^, while the CL−20/TNT cocrystal structure exhibits an absorption characteristic peak at 3107 cm^−1^. Based on the analysis of the infrared spectrum test results above, the characteristic peak frequency of the CL−20/TNT cocrystal has changed slightly compared to that of the pure component due to different intermolecular forces, indirectly confirming that CL−20/TNT explosive cocrystal can be synthesized by using the opposite spray method with aerodynamic atomized droplets.

### 3.5. Testing Analysis of X-ray Diffraction

The diffraction peaks of the CL−20/TNT samples prepared by the opposite spray method are located at 8.76°, 9.48°, 12.28°, 14.52°, 21.48°, 22.52°, 25.00°, 28.92°, etc., as shown in Figure 6c. Compared with the diffraction peak positions of the two raw materials, the peaks at 8.76° and 9.48° were not reflected in the raw material diffraction spectra, but it can be seen from Figure 6c that they were generated in the cocrystal sample. From analysis of Figure 6c, the diffraction peaks of the CL−20/TNT cocrystal sample have certain deviations at 12.28°, 14.52°, etc. Figure 6a shows that the diffraction peaks of CL−20 are at 15.77°, 16.43°, and 30.41°, while Figure 6b indicates that the characteristic peaks of TNT at 17.75°, 29.80°, and 33.50° have also disappeared. From the diffraction spectrum test analysis results, it is inferred that the sample can be determined as CL−20/TNT cocrystal, different from the single-component explosives.

### 3.6. Analysis of Binding Energy between Molecular Crystal Planes

By comparing the binding energy calculation results of different molecular models in Table 1, Table 2 and Table 3, it can be concluded that the binding energy of the CL−20/TNT cocrystal structure is higher than that of the CL−20 and TNT single components at 298 K, 318 K, and 338 K. This is because the CL−20 and TNT molecules are connected through intermolecular C−H−O interactions and nitro-benzene ring interactions in the cocrystal structure. Therefore, the regular and orderly arrangement of molecular crystals makes the cocrystal structure more stable, indicating that the thermodynamic stability of the cocrystal system is relatively good. At the same time, it was also shown that the binding energy between CL−20 and TNT increased first and then decreased with increasing temperature. This phenomenon indicates that there is an optimal temperature range for the preparation process of CL−20/TNT cocrystal, which is consistent with the previous result of using a cocrystal preparation process temperature of 30 °C. This also indicates that the results of the molecular dynamic calculations have certain theoretical guidance value for practical preparation processes. According to Table 5 and Table 6, the order of binding energies between CL−20 and TNT is: Eb (1 1 −1) > Eb (0 0 2) > Eb (1 1 0) > Eb (0 1 1) > Eb (1 0 1) > Eb (1 0 −1), while the order of binding energies between CL−20 and CL−20 is: Eb (0 0 2) > Eb (1 1 0) > Eb (0 1 1) > Eb (1 0 −1) > Eb (1 1 −1) > Eb (1 0 1). It can also be inferred that the binding energy between CL−20 and TNT is higher in the (0 0 2) and (1 1 −1) crystal planes, resulting in the lowest system energy and greater stability, which is conducive to the formation of cocrystal.

### 3.7. Cocrystal Mechanism Analysis of CL−20/TNT Based on Radial Distribution Function

The interaction between H atoms in TNT and O atoms in CL−20 is calculated in Figure 7a. The first peak between 2.5 and 3.0 Å belongs to hydrogen bonding, with a peak position of 2.73 Å. The second peak between 3.2 and 4.0 Å belongs to strong van der Waals interaction, and the peak intensities of the two peaks are relatively similar. Therefore, the strengths of hydrogen bonding and strong van der Waals force between H atoms and O atoms in the cocrystal structure are similar. The calculation results of the interaction between H atoms in CL−20 and O atoms in TNT are shown in Figure 7b. There is an obvious peak between 2.0 and 3.0 Å, indicating strong hydrogen bonding between them. The hydrogen bonding peak is located at 2.63 Å, and there is also a van der Waals interaction. However, it is obvious that the peak strength between 2.0 and 3.0 Å is larger, that is, the strength of hydrogen bonding between H atoms and O atoms is greater than that of van der Waals force. Figure 7c shows the interaction between the H atom in TNT and the N atom in CL−20, which is essentially a van der Waals interaction.

The calculation results indicate that, in the CL−20/TNT cocrystal structure, there are hydrogen bonds and strong van der Waals interactions between the H atom in TNT and the O atom in CL−20. There is a strong hydrogen bond between the H atom in CL−20 and the O atom in TNT, and the main van der Waals interaction between the H atom in TNT and the N atom in CL−20 is the hydrogen bond generated by the interaction between the H atom and the O atom in the cocrystal structure, which is consistent with the cocrystal mechanism formed by CL−20/TNT cocrystal explosives. The test results of the mechanical sensitivity of the CL−20/TNT explosive cocrystal are shown in Table 6.

### 3.8. Testing Analysis of Mechanical Sensitivity

According to Table 9, the impact sensitivity value of the CL−20/TNT cocrystal explosive prepared by the spray method is H_50_ = 36 cm. Compared with CL−20 (H_50_ = 15 cm), cocrystal significantly reduces the impact sensitivity of CL−20. The explosion probability of the friction sensitivity of the CL−20/TNT cocrystal explosive is 48%, which significantly reduces the friction sensitivity of the raw material CL−20 explosive by 100%. The combination of insensitive TNT and highly sensitive CL−20 through cocrystal technology forms a cocrystal at the molecular scale. Compared with traditional methods of coating and reducing sensitivity, this changes the internal composition and crystal structure of the explosives. In addition, due to intermolecular hydrogen bonding, on the one hand, it increases the stability of the cocrystal explosive molecular system, and on the other hand, it improves the impact resistance of the cocrystal molecules to mechanical external forces, resulting in a significant desensitization effect. It is worth noting that the particle size of ultrafine CL−20/TNT cocrystal explosives prepared by the opposite spray method ranges from 1–10 μm. Compared with the cocrystal explosive particles prepared by solvent evaporation method, the CL−20/TNT explosive prepared by the opposite spray method has a larger specific surface area of cocrystal grain size, which is beneficial for heat dissipation and less likely to generate hot spots locally. Therefore, the particle size can to some extent reduce the mechanical sensitivity of cocrystal explosives. Therefore, cocrystal technology can effectively reduce the sensitivity of highly sensitive explosives and improve their safety performance.

### 3.9. Analysis of Preparation Process Conditions for the Opposite Spray Method

During the experimental process, in order to improve the operability of the experimental preparation process, it is necessary to consider the effects of opposite spraying speed, nozzle spacing, solution concentration, aerodynamic pressure, and nozzle structure on cocrystal efficiency.

Opposite spraying speed is one of the influencing factors in the cocrystal preparation experiment. If the spraying speed is too low, the relative movement speed of the spray on both sides will decrease, and the collision force will be insufficient, which is not conducive to obtaining fine cocrystal particles. Moreover, the low spraying speed of the co-solution may cause some of the cocrystal solution droplets to fail to contact the antisolvent droplets, which will affect cocrystal efficiency. However, if the spraying speed is too high, the relative motion speed of the pneumatic atomized droplets will increase, and they may not fully contact and interact with the antisolvent droplets before being carried away by the airflow, which can also lead to a decrease in cocrystal efficiency.

Nozzle spacing also affects the preparation process of explosive cocrystal. If the nozzle spacing is too small, the contact time of the two strands of spray will be short, and some of the solution droplets will interact insufficiently and spray near the opposite nozzle, which will cause nozzle blockage. However, if the nozzle spacing is too large, the contact area of the two strands of spray will be smaller, weakening the interaction of the contact area, and some of the solution droplets will even fail to contact the antisolvent, which will affect cocrystal efficiency.

The concentration of the solution is also one of the important factors affecting the cocrystal preparation process. The concentration of the co-solution of explosives should not be too high. If it exceeds a certain limit, some of the co-solutions will be washed away by high-speed airflow without interaction in the collision area due to insufficient antisolvent extraction and crystallization ability, leading to a decrease in efficiency.

Aerodynamic pressure and nozzle structure have a significant impact on the feasibility and efficiency of explosive cocrystal synthesis. In practice, the spraying speed depends on the aerodynamic pressure and nozzle structure. When the nozzle structure is fixed, the spray velocity amplifies with an increase in aerodynamic pressure; when the pneumatic pressure is constant, the spray velocity increases with a decrease in the bore diameter of nozzle outlet.

Compared with the solvent evaporation method, the opposite spray method uses less solvent, and the anti-solvent is industrial water, which can be used within the room temperature range. This preparation method not only has low cost, but also high process safety. In addition, compared with the mechanical milling method, there is no mechanical stimulation of CL−20 or TNT during the preparation process of the spray method, which not only has high process safety, but also produces cocrystal explosives with small grain size and high production efficiency.

Of course, there are also some disadvantages to the opposite spray method used in this paper; for example, the size of the spray droplets caused by the spray speed and pressure is difficult to control, which easily leads to poor consistency of the prepared explosives cocrystal. Under conditions where the spray angle is not well controlled, there may also be a large number of droplets that cannot effectively collide with water solvent droplets, making it difficult to form cocrystal and potentially obtaining mixed crystals. These disadvantages may lead to a lower product yield rate of CL−20/TNT explosive cocrystal prepared by the opposite spray method.

The preparation of ultrafine CL−20/TNT cocrystal explosive by the opposite spray method should be carried out in a suitable explosion-proof laboratory with human–machine isolation. The key measures adopted include the following: (1) the nozzle and spray tank are located in the laboratory, and the experimenter controls the nozzle speed in the control room; (2) the process temperature of the spray technology can be controlled within the room temperature range; (3) the residual organic solvent and anti-solvent water mixture can be collected and uniformly recovered for treatment; (4) when collecting ultrafine CL−20/TNT cocrystals, the experimenter should wear a mask, eye mask, anti-static clothing, and rubber shoes.

## 4. Materials and Methods

### 4.1. Preparation Principle and Method of CL−20/TNT Cocrystal Explosive

#### 4.1.1. Principle of Opposite Spray Method with Pneumatic Atomized Droplets

The principle of the opposite spray method with pneumatic atomized droplets is to accelerate the explosive co-solution and antisolvent through the nozzle by the action of compressed gas to form high speed co-solution and antisolvent spray droplets. The two kinds of pneumatic atomized droplets continue to move in opposite directions under the effect of inertia and collide instantaneously in the middle area of the spray chamber with full contact. Due to the differences in solubility between CL−20 and TNT explosives in solvent and antisolvent, they reach a supersaturated state until small crystalline particles are rapidly precipitated. The small crystalline grains gradually lose their kinetic energy due to reverse frictional resistance and then fall to the bottom of the opposite spray chamber because of their own gravity. After the co-solution of the explosive is sprayed, the crystalline grains of the cocrystal explosive are obtained. The preparation principle of the opposite spray method with pneumatic atomized droplets is shown in Figure 8. It should be noted that changing the antisolvent to an atomized droplet form results in a relatively small amount of antisolvent in contact with the co-solution at a given moment. Therefore, it is necessary to control the continuous spraying of antisolvent and create an atmosphere filled with antisolvent in the spray chamber. At the same time, the explosive co-solution is intermittently sprayed to ensure that the sprayed explosive effectively reaches a supersaturated state.

Compared with the spray crystallization method, the opposite spray method with pneumatic atomized droplets also has the characteristics of atomization crystallization and rapid crystallization. Due to further atomization of the antisolvent, the contact area between the co-solution and the antisolvent is increased, resulting in an increase in the amount of supersaturated explosive and an acceleration of crystal precipitation rate. At the same time, the speed of the two spray solutions becomes higher, and the impact strength of the grains between each other is greater, which is conducive to obtaining ultrafine explosive cocrystal grains. Spray drying technology [24] can make the CL−20/TNT co-solution form ultrafine cocrystal particles, which can produce cocrystal explosives smaller than 1 μm that agglomerate into polycrystalline microspheres with a particle size of 1–10 μm. However, spray drying technology uses a process temperature as high as 65℃, making the spray process a typical thermodynamic coupling process. The cocrystal preparation of CL−20/TNT under thermal composite stimulation poses potential process hazards. In addition, this technology is discontinuous and has low yield, making it feasible for laboratory-level preparation research. However, there are bottlenecks when it comes to industrial production. In addition, CL−20/T NT cocrystal explosives with a particle size of around 270 μm were prepared by the solution cocrystallization method in reference [44], but the size of the prepared cocrystal explosives was relatively large. The opposite spray technology used in this paper used for the preparation of CL−20/TNT can be continuously produced and carried out at room temperature, and can produce ultrafine explosive cocrystals below 10 μm in size. Compared with the preparation methods proposed in references [24] and [44], our method has the same advantages of these other two preparation methods, while ensuring safety.

#### 4.1.2. Experimental Process for Preparing CL−20/TNT Cocrystal Explosive

The preparation method based on opposite spraying technology with pneumatic atomized droplets follows the following steps for the experimental process:

Based on the solubility characteristics of the two explosives, a suitable solvent and antisolvent are selected. The two explosive components are weighed according to a certain molar ratio and dissolved in a beaker containing a certain amount of solvent. Finally, they are completely dissolved to obtain a co-solution of explosives.

The spray drum is placed on the horizontal test bench, and then the explosive co-solution and antisolvent are respectively moved to the spray nozzle to prepare for the experiment.

Under the action of compressed gas, the antisolvent spray is formed continuously through the nozzle at one end of the opposite spray chamber. Continuous co-solution spray is formed at the opposite end and fully contacted in the middle area. The small grains that precipitate fall on the bottom of the cavity.

After the explosive co-solution is sprayed, the product is filtered and separated and then placed in an oven for a period. After the sample dries, it is taken out.

During the experimental process, it is necessary to consider the effects of injection speed, nozzle spacing, solution concentration, and tilt angle on the cocrystal effect. Considering these factors comprehensively, the operability, yield, and uniformity of the cocrystal explosive preparation experiment will be improved.

### 4.2. Experimental Chemicals and Instrumentation

The chemicals and reagents used in this experiment and their main properties are shown in Table 7.

Conventional air compressor instruments and various types of testing equipment were applied in the experiment. The main instruments and their models used in this experiment are shown in Table 8.

### 4.3. Preparation of CL−20/TNT Cocrystal Explosive

#### 4.3.1. Selection of Experimental Conditions

Based on the opposite spray method with pneumatic atomized droplets, it is necessary to determine the opposite spraying speed, nozzle spacing, and solution concentration before the experimental process begins. A small number of chemicals and reagents are used for preliminary experiments to explore reasonable experimental conditions and provide a certain basis for further experiments.

After preliminary exploration experiments, in the preparation process of CL−20/TNT cocrystal explosives, acetone was selected as the solvent for this experiment, water was selected as the counter solvent, and a feed ratio of 1:1 was used for the experiment. The spraying speed of the explosive co-solution is determined to be 10 mL/min, and the spraying speed of the antisolvent water is determined to be 60 mL/min. The distance between the two nozzles is 25 cm, and the concentration of explosive co-solution between CL−20 and TNT is about 0.10 mmol/mL. Experimental preparation of CL−20/TNT cocrystal explosives was carried out to investigate the feasibility of pneumatic atomization on the preparation of cocrystal explosives using the opposite spray method.

#### 4.3.2. Experimental Steps for Preparing CL−20/TNT Cocrystal Explosive

The experimental steps for preparing the CL−20/TNT cocrystal explosive using the opposite spray method with pneumatic atomized droplets are as follows:

According to the principle of a molar ratio of CL−20 and TNT of 1:1, the CL−20 and TNT explosives are weighed and poured into a beaker. And then a certain amount of acetone solvent is measured and transferred into a beaker.

The temperature of the ultrasonic cleaner is set to 30℃, and the beaker containing the explosive solution is placed in the ultrasonic cleaner. The explosive solution is thoroughly sonicated for 20-30 min until completely dissolved. Then the solution is filtered to obtain a total concentration of approximately 0.17 mmol/mL of CL−20 and TNT as a co-solution of explosives.

The support frame is placed on a horizontal experimental platform, and the spray drum is placed smoothly on the support frame. The explosive co-solution is transferred to the glass spray, and the antisolvent water is added to the spray bottle and placed at the spray ports at both ends of the counter spray drum.

Under the action of 0.1 MPa compressed gas, water spray is formed continuously at 60 mL/min at one end of the spray drum after being accelerated by the nozzle. At the other end, the spray of co-solution is continuously formed at a speed of 10 mL/min, which makes it collide in the middle area of the chamber, and the small grains are precipitated until the explosive co-solution in the spray is completely sprayed.

After a few minutes, the experimenter ensures that the co-solution droplets were in full contact with the antisolvent droplets and allowed the explosive molecule crystals to completely precipitate.

The product grains and water at the bottom of the spray drum are collected into a beaker, and the product is filtered and separated using a vacuum pump. The product is placed in an oven for 2 h, and the sample is taken out after drying at 60℃.

### 4.4. Molecular Model Construction of CL−20/TNT Cocrystal Explosive

#### 4.4.1. Molecular Crystal Plane Model of the CL−20 Explosive

Original crystal cells of CL−20 and TNT were imported, and the amorphous cell module was used to establish amorphous crystal cells of CL−20 and TNT with densities set at 2.044 g/cm^3^ and 1.654 g/cm^3^, respectively. Using the Morphology module to calculate the growth crystal planes of CL−20 in vacuum, relevant information on the six main growth crystal planes of CL−20 was obtained, as shown in Table 9. Six crystal plane models of CL−20 were obtained by cutting the CL−20 supercell, as shown in Figure 9.

#### 4.4.2. Molecular Model of the CL−20/TNT Cocrystal Explosive

Firstly, the crystal plane model of CL−20 was used as the first layer, and the crystal cell model of TNT was used as the second layer. The layer models of CL−20 crystal plane and TNT were established. Then, with the crystal plane model of CL−20 as the first layer and the cell model of CL−20 as the second layer, the crystal plane of CL−20 and the layer model of CL−20 were constructed. Finally, the first layers of these two molecular models were respectively constrained.

After minimizing the energy of the layered model, the NVT ensemble MD simulation was performed under the Compass force field with a temperature set at 298 K. The Andersen temperature control method was selected, with a time step of 1 fs and a simulation time of 200 ps. The simulation process was calculated using the atom-based and Ewald methods for van der Waals and electrostatic simulation, respectively. The structure with the lowest energy was selected as the final equilibrium structure between the CL−20 crystal plane and TNT, and the CL−20 crystal plane and CL−20, as shown in Figure 10 and Figure 11.

## 5. Conclusions

It is proposed to use the opposite spray method with pneumatic atomized droplets to prepare explosive cocrystal in this paper. Afterwards, the preparation principle of the opposite spray method was explained, and the influencing factors during the experimental process were analyzed. The experimental results have validated the feasibility of preparing CL−20/TNT cocrystal using the opposite spray method. Some conclusions can be drawn, as follows:

The opposite spray method is to use a nozzle to form spray droplets of explosive co-solution and antisolvent at the same time, and then to make the droplets fully contact and obtain cocrystal sample grains after reaching supersaturation. The solution concentration, nozzle spacing, aerodynamic pressure, nozzle structure, and spraying speed during the preparation process of cocrystal explosives can all affect the size of the cocrystal grains and synthesis efficiency.

Based on the opposite spray method with pneumatic atomized droplets, is the single components are dissolved in acetone to form a co-solution according to a ratio of TNT to CL−20 of 1:1. Under conditions of 0.1 MPa pneumatic pressure and a 1 mm bore nozzle diameter at 30℃, the spraying water antisolvent is continuously sprayed at one end of the spray drum, while the co-solution is intermittently sprayed at the other end, and full contact is made in the middle part to obtain CL−20/TNT cocrystal grain samples.

The molecular dynamics calculation method was used to calculate the binding energy and radial distribution function between CL−20 and TNT explosives. It was found that there is a strong binding performance between CL−20 and TNT molecules, and the order of binding energy between each crystal plane was given. Meanwhile, the influence of temperature on their binding energy was analyzed. There is a strong hydrogen bonding interaction between the H atom in CL−20 and the O atom in TNT, which is the mechanism of cocrystal formation between CL−20 and TNT.

The prepared cocrystal samples were characterized by scanning electron microscopy images, differential thermal testing, infrared spectroscopy, and diffraction patterns, indicating that the opposite spray method can prepare explosive cocrystal. By using this method, a superfine explosive cocrystal can be obtained, which is approximately 1-10 μm and has a uniform particle size distribution compared to commonly used methods for preparing cocrystal. Therefore, the mechanical sensitivity of the ultrafine CL−20/TNT explosive cocrystal was significantly reduced. However, the obtained fine cocrystal grains are prone to aggregation, and further research is needed to determine the exact reasons for their aggregation.

## Figures and Tables

**Figure 1 ijms-25-09501-f001:**
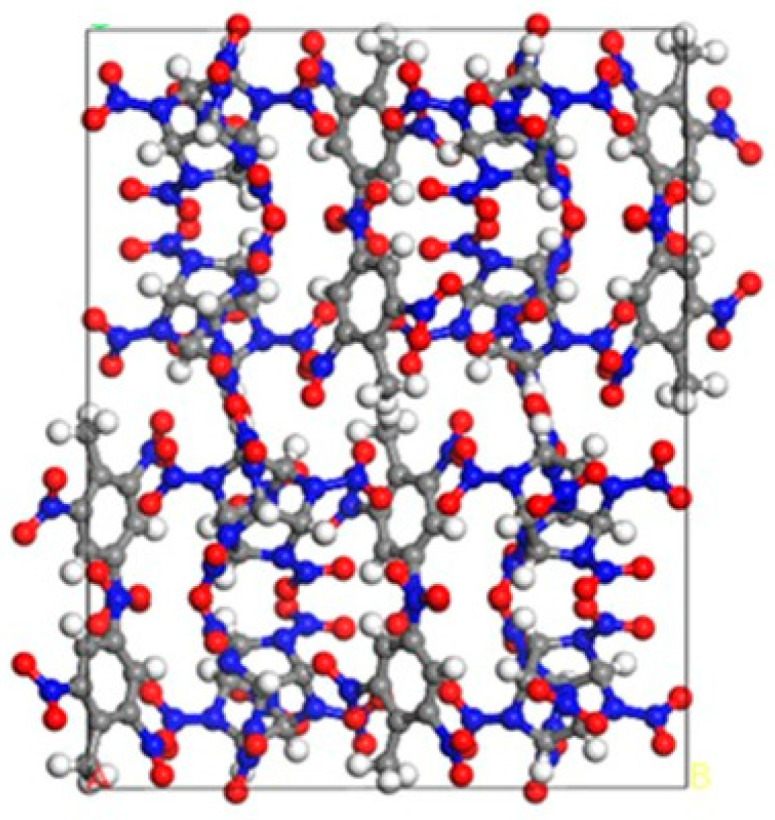
Cocrystal structure of CL−20/TNT.

**Figure 2 ijms-25-09501-f002:**
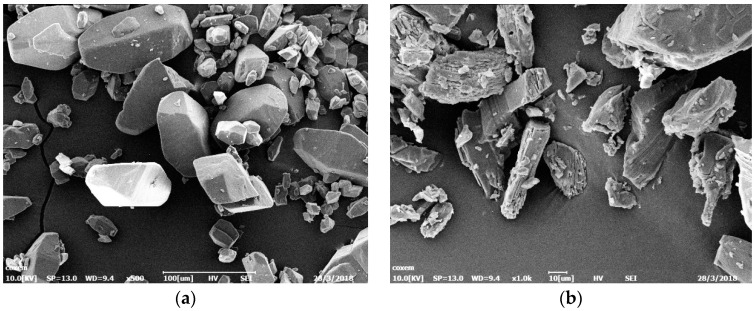
Scanning electron microscopy images of the raw explosive materials CL−20 and TNT and the cocrystal explosive CL−20/TNT. (**a**) CL−20 with SEM at 500 times magnification; (**b**) TNT with SEM at 1000 times magnification; (**c**) CL−20/TNT with SEM at 5000 times magnification; and (**d**) CL−20/TNT with SEM at 10,000 times magnification.

**Figure 3 ijms-25-09501-f003:**
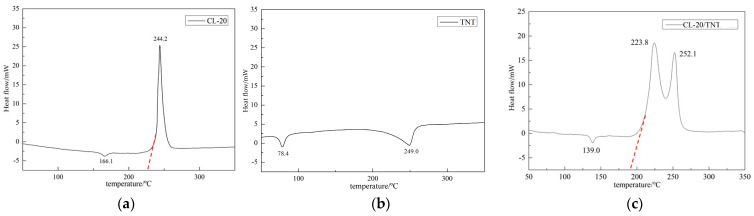
DSC curve test results of the samples. (**a**) Raw CL−20; (**b**) raw TNT; and (**c**) CL−20/TNT.

**Figure 4 ijms-25-09501-f004:**
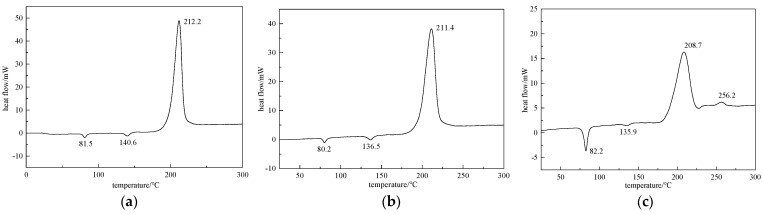
DSC curve test results of the cocrystals with different feeding ratios. (**a**) 1:2; (**b**) 1:3; and (**c**) 1:4.

**Figure 5 ijms-25-09501-f005:**
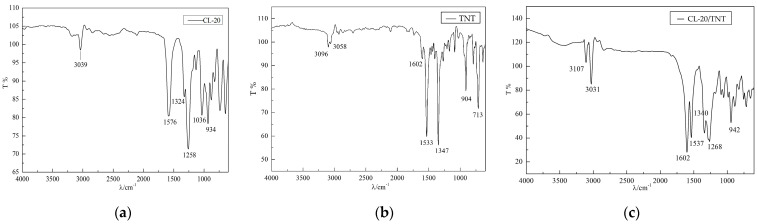
Infrared spectral curve image of the sample. (**a**) Raw CL−20; (**b**) raw TNT; and (**c**) CL−20/TNT.

**Figure 6 ijms-25-09501-f006:**
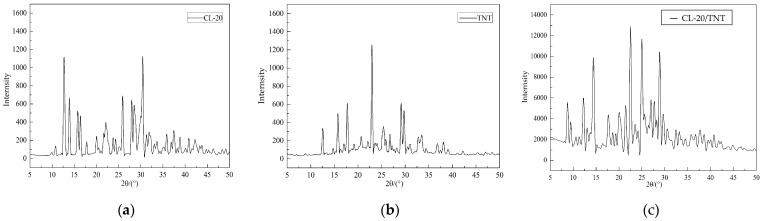
X−ray diffraction curve images of the samples. (**a**) raw CL−20; (**b**) raw TNT; and (**c**) CL−20/TNT.

**Figure 7 ijms-25-09501-f007:**
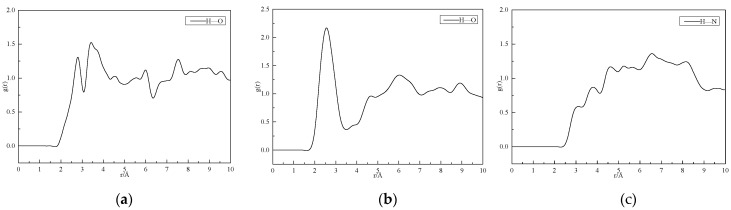
Radial distribution function of CL−20/TNT cocrystal structure. (**a**) TNT(H)—CL−20(O); (**b**) CL−20(H)—TNT(O); and (**c**) TNT(H)—CL−20(N).

**Figure 8 ijms-25-09501-f008:**
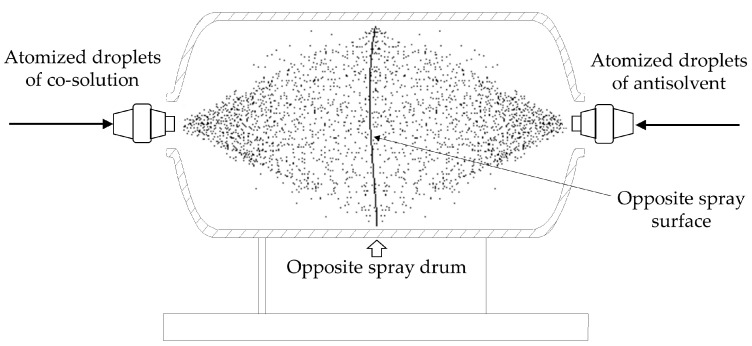
Principal schematic of the opposite spray method with pneumatic atomized droplets.

**Figure 9 ijms-25-09501-f009:**
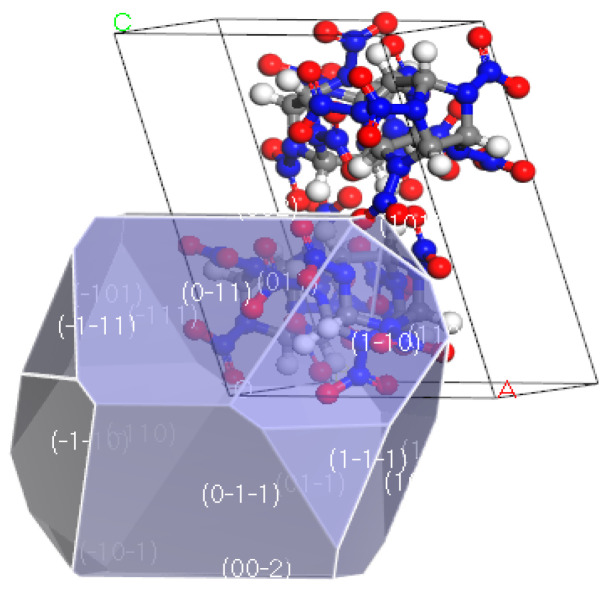
The growth crystal form of CL−20.

**Figure 10 ijms-25-09501-f010:**
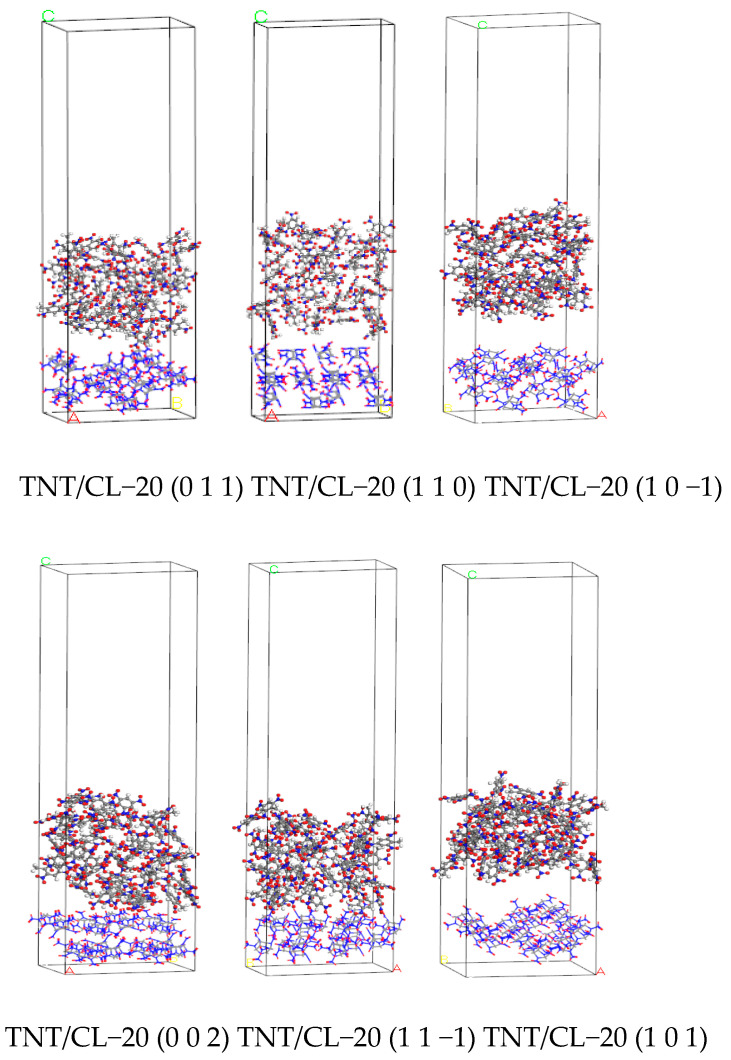
The equilibrium structure of CL−20 and TNT crystal plane.

**Figure 11 ijms-25-09501-f011:**
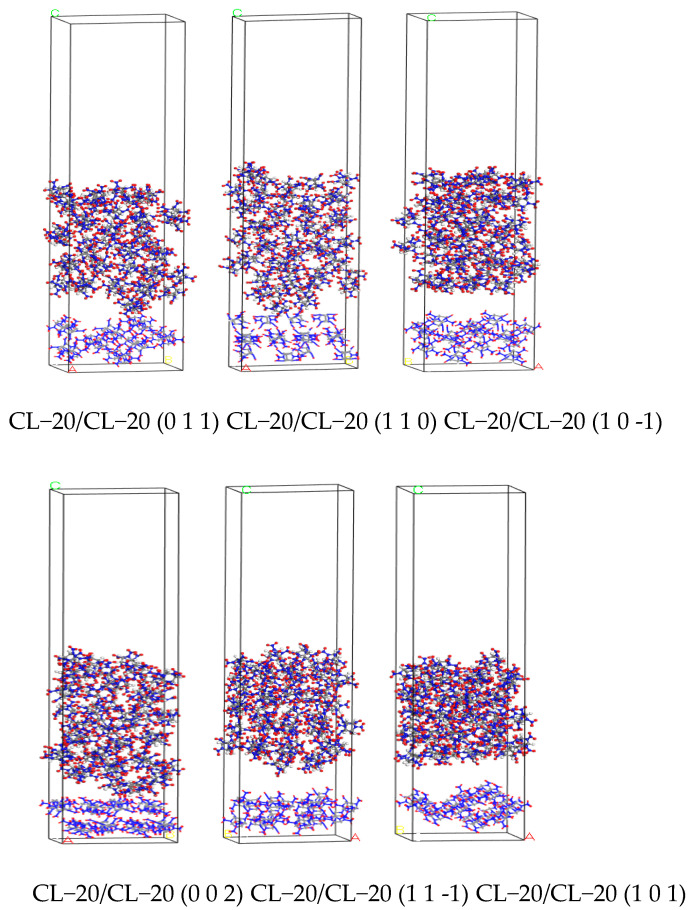
The equilibrium structure between CL−20 crystal planes.

**Table 1 ijms-25-09501-t001:** Binding energy of the CL−20/TNT cocrystal and single components at 298 K.

Sample	E_total_ /(kcal/mol)	E_layer (1)_ /(kcal/mol)	E_layer (2)_ /(kcal/mol)	E_bind_ /(kcal/mol)
CL−20/TNT	−7424.058	1925.070	−9248.818	100.310
CL−20/CL−20	−18,489.851	−9266.570	−9215.049	8.232
TNT/TNT	2485.858	1220.582	1269.425	4.149

**Table 2 ijms-25-09501-t002:** Binding energy of the CL−20/TNT cocrystal and single components at 318 K.

Sample	E_total_ /(kcal/mol)	E_layer (1)_ /(kcal/mol)	E_layer (2)_ /(kcal/mol)	E_bind_ /(kcal/mol)
CL−20/TNT	−7454.284	1893.402	−9162.901	184.785
CL−20/CL−20	−1,8434.556	−9196.321	−9223.500	14.735
TNT/TNT	2464.076	1277.202	1308.020	121.146

**Table 3 ijms-25-09501-t003:** Binding energy of the CL−20/TNT cocrystal and single components at 338 K.

Sample	E_total_ /(kcal/mol)	E_layer (1)_ /(kcal/mol)	E_layer (2)_ /(kcal/mol)	E_bind_ /(kcal/mol)
CL−20/TNT	−7411.435	1950.566	−9202.705	159.296
CL−20/CL−20	−18,492.884	−9219.501	−9263.576	9.807
TNT/TNT	2475.853	1272.912	1297.089	94.148

**Table 4 ijms-25-09501-t004:** Binding energy of various crystal planes between CL−20 and TNT.

Crystal face	E_total_ /(kcal/mol)	E_layer (1)_ /(kcal/mol)	E_layer (2)_ /(kcal/mol)	E_bind_ /(kcal/mol)
0 1 1	−2819.4	−4620.6	1820.8	19.6
1 1 0	−2681.4	−4495.6	1837.1	22.9
1 0 −1	−2930.2	−4786.3	1860.9	4.8
0 0 2	−3104.7	−4858.1	1842.2	88.8
1 1 −1	−2523.3	−4303.4	1877.4	97.3
1 0 1	−2861.9	−4741.3	1884.6	5.2

**Table 5 ijms-25-09501-t005:** Binding energy of various crystal planes between CL−20 and CL−20.

Crystal face	E_total_ /(kcal/mol)	E_layer (1)_ /(kcal/mol)	E_layer (2)_ /(kcal/mol)	E_bind_ /(kcal/mol)
0 1 1	−18,541.5	−4620.5	−13,895.1	25.9
1 1 0	−18,521.4	−4495.4	−13,996.9	29.1
1 0 −1	−18,726.8	−4786.1	−13,933.4	7.3
0 0 2	−18,824.9	−4857.9	−13,916.8	50.2
1 1 −1	−18,230.4	−4303.3	−13,920.1	7.0
1 0 1	−18,652.8	−4741.2	−13,905.5	6.1

**Table 6 ijms-25-09501-t006:** Mechanical sensitivity test results of samples.

Sample	Impact Sensitivity H_50_/cm	Friction Sensitivity P/%
Raw CL−20	15	100
Raw TNT	102	6
CL−20/TNT	36	48

**Table 7 ijms-25-09501-t007:** Chemicals, reagents, and their main properties.

Name	Specifications	Properties
CL−20	-	White powder, easily soluble in acetone, methanol, acetonitrile, and ethyl acetate, slightly soluble in ethanol, insoluble in chlorinated hydrocarbons and water.
TNT	-	Yellow flake or powder, highly soluble in acetone, toluene, benzene, and chloroform, slightly soluble in ethanol and carbon tetrachloride, insoluble in water.
Acetone	analytical pure	Colorless transparent liquid, flammable and volatile.

**Table 8 ijms-25-09501-t008:** Main instruments and their models used in the experiment.

Name	Model	Manufacturer
Electronic balance	AL204-IC	Mettler Toledo Instruments Co., Ltd., Shanghai, China.
Ultrasonic cleaner	KQ-250DE	Kunshan Ultrasonic Instrument Co., Ltd., Kunshan, China.
Air compressor	600W-30L	Shanghai Shengxi Hardware and Electromechanical Co., Ltd., Shanghai, China.
Opposite spray drum	5 L	Anqing Shengxing Plastic Industry, Anqing, China.
Circulating water multi-purpose vacuum pump	SHB-IIIS	Zhengzhou Changcheng Science and Technology Industry and Trade Co., Ltd., Zhengzhou, China.
Thermostatic incubator	SPX-150B-D	Shanghai Boxun Industrial Medical Equipment Factory, Shanghai, China
Scanning electron	EM-30	COXEM Company, Datian city, Korea
DSC	HCT-1	Beijing Hengjiu Technology Co., Ltd., Beijing, China.
Infrared spectrometer	Perkin Elmer Spectrum 100	Platinum Elmer Inc., MA, USA
X-ray diffractometer	DX-2700	Dandong Haoyuan Instrument Co., Ltd., Dandong, China.
Impact sensitivity device	WL-1	Shanxi Institute of Applied Physics and Chemistry, Xi’an, China
Friction sensitivity device	MGY-I	Shanxi Institute of Applied Physics and Chemistry, Xi’an, China

**Table 9 ijms-25-09501-t009:** Growth crystal plane of CL−20 in vacuum.

hkl	Multiplicity	d_hkl_	Surface Area	E_att_(Total)	Distance	Area Ratio
(0 1 1)	4	8.909	156.626	−83.975	83.975	38.914
(1 0 −1)	2	8.180	170.570	−85.080	85.080	13.749
(1 1 0)	4	6.981	199.874	−89.337	89.337	26.114
(1 1 −1)	4	6.841	203.976	−94.024	94.024	8.421
(0 0 2)	2	6.364	109.634	−90.810	90.810	10.713
(1 0 1)	2	6.251	223.204	−109.305	109.305	2.090

## Data Availability

All data are contained within the article.

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
