# Peer review of "Preparation and Molecular Dynamic Simulation of Superfine CL−20/TNT Cocrystal Based on the Opposite Spray Method"

_ijms, 2024, doi:10.3390/ijms25179501_

Round 1
Reviewer 1 Report
Comments and Suggestions for Authors
The author of the paper "Preparation and Molecular Dynamic Simulation of Superfine CL-20/TNT Cocrystal based on the Opposite Spray Method" investigates a new method to prepare ultrafine CL-20/TNT cocrystals using an opposite spray technique. The study addresses issues like slow crystallization rates, varying grain sizes, complex preparation processes, and safety concerns associated with traditional methods. By optimizing the process conditions and employing molecular dynamic simulations, the author aims to improve the crystallization rate, grain size uniformity, and safety of CL-20/TNT cocrystals. Key findings include the identification of optimal solvent (acetone), spray temperature (30℃), and feed ratio (1:1) for the formation and growth of cocrystals, which resulted in particles approximately 10 μm in size. However, there are some aspects where revisions are recommended:
1. The description of the opposite spray method could be more detailed. How does this method compare with other traditional methods in terms of efficiency and quality of the produced cocrystals?
2. While thermal stability is briefly mentioned, a comprehensive thermal analysis comparing the cocrystals with pure CL-20 and TNT is missing. How do the thermal decomposition temperatures and heats of reaction compare?
3. The particle size distribution of the cocrystals is mentioned, but the impact of particle size on the performance and stability of the cocrystals is not thoroughly discussed. Could the author expand on this aspect?
4. Given the hazardous nature of CL-20 and TNT, the environmental and safety implications of the opposite spray method should be discussed. What measures are taken to ensure safe handling and minimal environmental impact? Providing a safety assessment would be prudent.
5. A comparative analysis with other preparation methods and types of cocrystals would provide a broader context for the significance of the study’s findings. How does the performance of these superfine cocrystals compare with those prepared by methods like solvent evaporation or mechanical milling?
Comments on the Quality of English Language
Minor editing of English language required
Reviewer 2 Report
Comments and Suggestions for Authors
In this manuscript, the authors reported an opposite spray method to prepare cocrystal CL-20/TNT with superfine size of approximately 10μm, and the formation conditions were systematically investigated. Molecular dynamic simulation was adopted to explain the mechanism from the molecular perspective. The work is meaningful and provide inspiration for the preparation of insensitive and high performance ammunition. However, there are several issues need to be addressed before it can be considered for the publication in Int. J. Mol. Sci:
1. Please pay attention to references format, all authors should listed, eg: ref. 21, 22, 24, check all. Please also double check the format of authors, the surname abbreviation is not consistent, eg: ref. 12, 13, 14, 15, check all.
2. The superfine cocrystal CL-20/TNT was prepared with the feed ratio of molar 1:1, but I doubt this is the optimal ratio because of there is not perfect match between the two molecule. Can the author compare the performance of other ratio or provide a reasonable explain?
3. It is possible to obtain the cocrystal structure and determined by X-ray diffraction? It will beneficient to confirm the real molecular ratio of the cocryal composition.
Comments on the Quality of English LanguageNo
Reviewer 3 Report
Comments and Suggestions for Authors
The authors of the article “Preparation and Molecular Dynamic Simulation of Superfine CL-20/TNT Cocrystal Based on the Opposite Spray Method” have presented a unique method. This is an interesting contribution to the energetic materials community, and I recommend major revisions as outlined below.
-
The references in the introduction need to be corrected. The authors should remove the “Error! Reference source not found” text appearing in lines 45, 52, and 58.
-
In lines 42-43, the authors should cite recent reports on the synthesis of high-energy material co-crystallization and salt co-crystallization.
-
Does the density or morphology differ compared to cocrystals obtained by other methods? It would be interesting to know the effect of the “Opposite Spray Method” on the physicochemical properties.
-
Correct the typo occurring in several instances. On line 447, change “Oppoiste” to “opposite.”
-
While the authors have commented on the disadvantages of other methods in the introduction, they should also discuss the advantages and limitations of the presented method, including product yields, in the conclusions.
-
The authors conducted DSC experiments at a 10°C/min heating rate. What is the difference in thermal decomposition of the co-crystals compared to what is reported in the literature?
-
Most importantly, the impact and friction sensitivity properties of the material are not reported. The primary goal of making the CL-20/TNT co-crystal is to reduce the sensitivity of CL-20.
No comments
Round 2
Reviewer 3 Report
Comments and Suggestions for Authors
The Authors have made the suggested changes. Manuscript can be accepted.
Comments on the Quality of English LanguageNo Comments